# Cell-type Neural Ordinary Differential Equation Models for Parsing Biologically-Constrained Contributions to Neural Dynamics

## Abstract

Understanding how populations of individual neurons interact to shape the overall dynamics of neural activity is a central question in computational and systems neuroscience. Recent work has shown that neural ordinary differential equation (NODE) models are able to model neural activity dynamics with high accuracy and interpretability of the underlying dynamics. However, existing NODE models treat all neurons as part of a homogenous group, preventing understanding how underlying neural populations (e.g. cell types) contribute to the overall dynamics. Here, we introduce Cell-Type NODE (CT-NODE) models. These models A) decompose the overall dynamics into components specific to each population, allowing understanding each population's interactions with one another; and B) provide biological constraints on the contributions of excitatory and inhibitory populations towards the dynamics, using a variant of monotonic neural networks. Using both synthetic and recorded neural activity data during a naturalistic climbing task, we show that CT-NODE models can provide equivalent, or greater, accuracy of dynamics modeling compared to standard NODE models, while enabling a new-found biologically-constrained understanding of neural populations' interactions and roles in the underlying dynamics.

## 1 Introduction

Recent advances in large-scale neural recordings reveal rich, high-dimensional population activity that invites dynamical systems explanations (Jun et al., 2017; Stirman et al., 2016; Ahrens et al., 2013; Sauerbrei et al., 2020; Itokazu et al., 2018). Latent dynamical models capture these data via low-dimensional states whose temporal evolution generates observed activity, achieving strong denoising and alignment with behavior (Macke et al., 2011; Petreska et al., 2011; Pandarinath et al., 2018).

Neural ordinary differential equation models (Neural ODEs, or NODEs), which model the change of dynamics over time with flexible neural networks, have recently emerged as a promising approach towards modeling the latent dynamics of neural population activity. NODEs have been shown to have excellent accuracy, predicting neural activity data more accurately than recurrent neural network (RNN) models (Kim et al., 2021). They have also demonstrated accurate modeling of low-dimensional dynamics within a low-dimensional space, whereas RNN models require a greater dimensionality (due to their less flexible dynamics) to accurately capture low-dimensional dynamics (Sedler et al., 2023). NODEs also can provide a particularly interpretable view of the underlying dynamics, as flow fields can easily be drawn in the low-dimensional space demonstrating how neural activity evolves over time, as was recently done to discover neural underpinnings of decision commitment (Luo et al., 2025).

Despite great promise, there are two fundamental limitations of existing NODE models, especially relating to modeling widely emerging datasets with multiple recorded neural populations (e.g. brain regions and cell types). First, there has been great interest in understanding and modeling the roles of, and interactions between, different neural populations (Keeley et al., 2020; Perich et al., 2020;

Liu et al., 2025; Dowling & Savin; Cunningham & Yu, 2014; Li et al., 2024; Koukuntla et al., 2024; Jha et al., 2025; Gokcen et al., 2022; 2023). This is evidenced by recent work within other dynamics architectures, for example linear (Semedo et al., 2014), switching linear (Glaser et al., 2020), and switching nonlinear dynamical systems (Karniol-Tambour et al., 2022), to decompose the overall latent dynamics into the sum of the influences of different populations. It is not possible within current NODE models to understand how populations interact to drive the overall neural dynamics of the system. Second, biological neural circuits comprise distinct excitatory (E) and inhibitory (I) cell populations with sign-constrained interactions (Dale's law). Unlike dynamics architectures like RNNs (Song et al., 2016) and linear dynamical systems (Jha et al., 2024), current NODE models don't enable incorporating these biological constraints. This omission prevents constraining the dynamics of models (which are often poorly constrained (Prinz et al., 2004; Das & Fiete, 2020; Beiran & Litwin-Kumar, 2024) toward biological solutions, and obscures how distinct cell classes contribute to circuit computations.

To tackle these challenges, we developed cell-type-aware neural ordinary differential equation (CT-NODE) model dynamics. CT-NODE assigns separate latent subspaces to E and I populations and constrains cross-population influences in this latent space to respect Dale's law so that E (and I) neurons only excite (and inhibit). To do so, we develop a dynamics parameterization that builds upon monotonic neural networks (Daniels & Velikova, 2010), which enables the dynamics to retain the flexibility inherent to NODE models, while now respecting biological constraints. After formulating the model (Section 2), we evaluate our model on synthetic data generated from a winner-take all decision-making circuit (Section 3), and naturalistic mouse climbing data recorded in primary (M1) and secondary (M2) motor areas (Section 4). We demonstrate how CT-NODE models enable interpreting how individual populations contribute to the overall dynamics (e.g. within flow fields), and how they interact with each other over time, which is not possible within standard NODE models. Crucially, this added biological interpretability does not come at the cost of prediction performance versus competing models. Together, our findings suggest a path toward simple, accurate, and biologically grounded latent dynamics that elucidate computation in neural circuits.

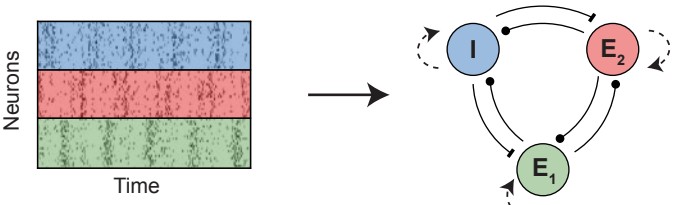

Figure 1: *CT-NODE Schematic*. (Left) Raster plots of neural firing rates over time for three example populations. (Right) A schematic of our CT-NODE model is trained to infer population interactions from neural activity for both E and I populations, with positive (dot), negative (line), and unconstrained (dashed arrow) interactions.

## 2 Cell-type Neural Ordinary Differential Equations (CT-NODE)

### 2.1 Background: Neural Ordinary Differential Equations (NODEs)

NODEs model the dynamics of a variable $z$ at time $t$ are learned according to:

$$z_t = z_{t-1} + \int_{t-1}^{t} f_\theta(z_{t-1}) \, dt \tag{1}$$

where $f_\theta$ is a neural network parameterizing the latent dynamics vector field, and can be interpreted as the rate of change of $z$ over time (its derivative). Critically, this formulation provides a skip connection from the previous state, $z_{t-1}$, focusing $f_\theta$'s full expressivity towards dynamics.

### 2.2 Background: NODE-based Sequential Autoencoders

Here, as in past work, (Sedler et al., 2023), we model the high-dimensional neural activity as having a lower-dimensional latent representation, and the NODE dynamics occur within this latent state.

This framework has been referred to as a NODE-based sequential autoencoder (Sedler et al., 2023), although we will here just refer to these models as NODEs for brevity.

Let $x_t \in \mathbb{R}^N$ denote the observed binned spike counts of $N$ neurons at time $t$. Predicted firing rates $\hat{y}_t$ are mapped from $D$ latents $z_t \in \mathbb{R}^D$ with weights $W_{\text{dec}}$ and biases $b_{\text{dec}}$, according to:

$$\hat{y}_t = \exp\!\big(W_{\text{dec}}\, z_t + b_{\text{dec}}\big), \qquad \hat{x}_t \in \mathbb{R}^N > 0. \tag{2}$$

Neural dynamics evolve in this lower dimensional state according to the dynamics specific in equation 1. The dynamics evolve in a deterministic matter from a learned initial latent state (the initial condition), $z_0$. For each sequence (e.g. trial) of data that is modeled, a separate initial condition is learned. We learn initial conditions by passing the high-dimensional neural activity through a Gated Recurrent Unit (GRU)-based encoder. More precisely, spike counts $x$ over time $T_{\text{enc}}$ are input into a GRU with learned weights $\phi$, and whose hidden state $h$ is linearly mapped to the initial latent $z_0$ using encoder weights $W_{\text{enc}}$ and biases $b_{\text{enc}}$.

$$h = \text{GRU}_\phi\big(x_{1:T_{\text{enc}}}\big), \qquad z_0 = W_{\text{enc}}\, h + b_{\text{enc}} \tag{3}$$

## 2.3 THE CT-NODE MODEL

To disentangle the roles of distinct cell types in a population, we modify the standard NODE model in two ways: (1) we define a distinct set of latent variables for the activity of each cell class; and (2) we constrain the dynamics to obey functional properties of these cell types. We refer to the resulting models as Cell-Type Neural Ordinary Differential Equations (CT-NODE) models.

### 2.3.1 CONSIDERING SINGLE POPULATIONS OF EXCITATORY AND INHIBITORY NEURONS

Let's first assume we have a single E and single I population. We use distinct sets of latents for E ($z^E$) and I ($z^I$), where each population can have multiple latents, $D_E$ and $D_I$ respectively. We can thus write out the entire latent state as:

$$z = \begin{bmatrix} z^E \\ z^I \end{bmatrix}, \qquad z^E \in \mathbb{R}^{D_E}, \ z^I \in \mathbb{R}^{D_I}, \ D_E + D_I = D \tag{4}$$

We model the change in dynamics (i.e., the latent vector field) as the sum of a term capturing population interactions plus a nonlinear self-term. The E population changes over time due to its intrinsic dynamics $f_{EE}(z^E)$ and its interactions with the I population $f_{EI}(z^I)$. Similarly, the I population changes over time due to its intrinsic dynamics $f_{II}(z^I)$ and its interactions with the E population $f_{IE}(z^E)$.

$$z_t^E = z_{t-1}^E + \int_{t-1}^{t} f_{EE}(z_{t-1}^E) + f_{EI}(z_{t-1}^I)\, dt,$$
$$z_t^I = z_{t-1}^I + \int_{t-1}^{t} f_{II}(z_{t-1}^I) + f_{IE}(z_{t-1}^E)\, dt \tag{5}$$

We take two steps to impose Dale's law constraints on the latent interactions for $f_{IE}$ and $f_{EI}$ – that is, to ensure that the influence from I latents is negative and E latents is positive. We'll initially describe this for E latents. First, for these functions, we use feedforward monotonic neural networks (Daniels & Velikova, 2010), where all the weights within the network are constrained to be nonnegative. Thus, an increasing input to $f_{IE}$ leads to an increasing output. This can be thought of as a nonlinear equivalent of a nonnegative weight in a linear mapping. Still, this alone does not provide ultimate interpretability, as if the E latent itself has a negative value, the output can also be negative (which is also a current limitation of applying Dale's law within linear dynamical systems). Thus, second, we pass the output of our monotonic neural network through a softplus function to ensure the output is positive. When modeling the I outputs, we simply take the negative of this approach, ensuring a monotonically decreasing function with negative output:

$$f_{IE} = \text{softplus}\big(\tilde{f}_{IE}\big), \qquad f_{EI} = -\,\text{softplus}\big(\tilde{f}_{EI}\big) \tag{6}$$

where $\tilde{f}$ are monotonic neural networks. Overall, this approach allows learning flexible nonlinear dynamics, as in classic NODE models, but while retaining relevant E/I constraints.

The intrinsic self-terms $f_{II}$ and $f_{EE}$ are left as unconstrained feedforward neural networks, similar to how diagonal terms have been left unconstrained in E/I constrained linear dynamical systems (Jha et al., 2024). This is important for allowing populations' activities to 'leak' back to a baseline level.

### 2.3.2 Considering multiple populations of excitatory and inhibitory neurons

We now extend our above framework to the more general scenario of multiple E and I populations. For example, if there are E and I populations from multiple brain regions, our objective is to characterize how all these populations interact within and across regions. In the general case, the dynamics of each population is the sum of the 'influences' from all other populations, along with its own intrinsic dynamics. For $K$ simultaneously recorded populations, we assign population-specific dynamics functions for the $j^{th}$ population with learned population interactions, $f_{jk}$, and intrinsic dynamics, $f_{jj}$, as:

$$z = \begin{bmatrix} z^1 \\ \vdots \\ z^j \end{bmatrix}, \qquad z^j \in \mathbb{R}^{D_j}, \tag{7}$$

$$z_t^j = z_{t-1}^j + \int_{t_{-1}}^t f_{jj}(z_{t-1}^j) + \sum_{k \neq j}^K f_{jk}(z_{t-1}^k))dt \tag{8}$$

The dynamics functions $f_{jk}$ are parameterized as in equation 6 to follow Dale's law, where source E influences are positive and monotonically increasing, and source I influences are negative and monotonically decreasing, while the self-term is unconstrained.

### 2.3.3 Mapping to the high-dimensional neural space

So that latent groups can be interpreted as corresponding to prespecified neural populations, we have specified latent groups (e.g., $z^E$ vs. $z^I$) be constrained to map onto their corresponding pre-specified neuron subsets (e.g. known E and I neurons). We can thus write the readout as:

$$\hat{X}_t = \exp(W_{\text{dec}} z_t + b_{\text{dec}}) \qquad W \leftarrow W \odot M_{\text{pop}} \tag{9}$$

where $M_{\text{pop}}$ is a block-diagonal binary mask mapping latents to their corresponding neurons. We note that in some experiments below, we relax this mask to test the ability of the model to learn neuron types without prespecifying them all.

To ensure that the Dale's law constraints remain present in the high-dimensional neural activity space - e.g. all E neurons (not just E latents) have positive influences, we constrain all weights of $W_{dec}$ to be nonnegative, as in past work (Jha et al., 2024).

We also note that we tested more flexible nonlinear readouts within our initial experiments on simulated data below, and saw similar performance - thus, we use a linear readout for simplicity.

### 2.4 Model Training

We minimize the negative log-likelihood (NLL) over $T$ time steps and $N$ neurons between predicted rates $\hat{Y} = [\hat{y}_1...\hat{y}_N]^T$, and observed firing activity $X = [x_1...x_N]^T$:

$$\mathcal{L}_\theta(X, \hat{Y}) = -\frac{1}{TN} \sum_{t=1}^T \sum_{n=1}^N \log \text{Poisson}(x_{t,n} \mid \hat{y}_{t,n})$$

We initially trained two types of models - those that used the continuous integral version of the model, as specified in equation 1, and also a discretized version where time steps are predicted discretely from the previous time step. When using this continuous version, we computed NODE trajectories using a Runge-Kutta solver (Tsitouras, 2011). As we saw equivalent performance between discrete and continuous versions while replicating the findings of Sedler et al. (2023), the results below are shown for the discretized version of the models (with $\Delta t = 0.1$), which has faster training.

$$z_t = z_{t-1} + \Delta t * f_\theta(z_{t-1}) \tag{10}$$

Models were optimized using backpropagation through time, with further details particular to experiments (e.g. batch size, learning rate, and architecture hyperparameters) described in Appendix A. Lastly, model training and hyperparameter sweeps were run both locally and on SLURM CPU clusters using the Python library Hydra (Tristram & Bradshaw, 2009).

## 2.5 MODEL COMPARISONS

To serve as comparisons for our CT-NODE model, we additionally compared to several approaches of modeling the dynamics: 1) standard nonlinear NODE dynamics, for the purpose of determining whether the cell-type-based dynamical constraints that facilitate interpretability are at the cost of performance accuracy; 2) unconstrained linear dynamics and 3) cell-type constrained linear dynamics within the NODE framework, with the purpose of determining the importance of nonlinear dynamics, as cell-type constraints have previously been implemented within linear models (Jha et al., 2024) (although not this NODE formulation); 4) standard RNN dynamics, as cell-type constraints have previously been implemented within RNNs (Song et al., 2016). Details of these models are described in Appendix A.

## 3 APPLICATION TO SIMULATED NEURAL DATA

To demonstrate the feasibility of our method, we first validated its ability to recover known circuit dynamics on synthetic data from a canonical decision-making circuit (Wang, 2002). Briefly, this dataset consists of two E populations (E1 and E2), each containing 240 neurons, and one I population containing 400 neurons. Together, these populations form a decision-making model that achieves winner-take all competition between the two E populations based on feedback from the I population (Fig. 2A). This is a biophysically realistic network model that produces spiking activity across these three populations based on their interactions.

## 3.1 MODEL FITTING DETAILS

In this task, models were trained to reconstruct spiking activity from two-second trials where either population E1 or E2 would exhibit a sustained increase in spiking activity (Winner-Take All; examples in Fig. 2A). A total of 80 trials were generated from the simulation and randomly split into a training and validation set, with 80% of the trials used for training and 20% for validation. For each trial, the spiking activity was binned into 20 millisecond (ms) bins, creating a trial sequence length of 100 bins. Each 100-bin trial was input into each model to obtain the initial latent condition before rolling out the dynamics and aiming to reconstruct the original sequence of neural activity. The latents were partitioned by population with one latent per population (two E latents and one I latent).

## 3.2 RESULTS: PREDICTION ACCURACY

We found that our CT-NODE model could make accurate predictions of the trajectories of neural activity over time (Fig. 2C and 2D). We quantitatively compared the CT-NODE dynamics model to several approaches of dynamics modeling (Fig. 2B; motivated and described in section 2.5 and Appendix A). The CT-NODE slightly outperformed the NODE model, validating that introducing dynamical constraints does not hinder reconstruction performance while also adding the benefit of interpretability. In fact, the further constraints slightly helped predictions on held-out data. The CT-NODE model also outperformed the Linear CT-NODE and Linear NODE models, which we expected given that the winner-take all mechanism uses nonlinear dynamics. We also found that all NODE models outperformed the RNN baseline, most likely due to the long prediction horizon of this task.

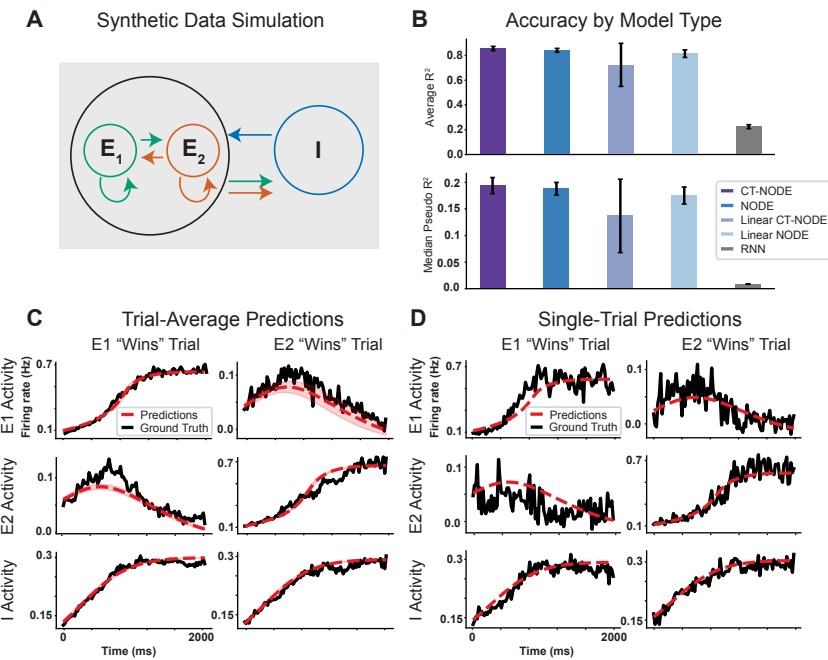

Figure 2: *Synthetic data predictions*. **A)** Schematic of the synthetic dataset Wang (2002), including two E populations and one I population. **B)** Future prediction accuracy is compared across model types. On top, the $R^2$ of the population averages. On bottom, the median pseudo-$R^2$ of individual neurons' spike count predictions. **C)** Trial-average predictions against ground truth firing rates for each population average. **D)** Example single-trial predictions for population averages.

## 3.3 RESULTS: DYNAMICS INTERPRETABILITY

The CT-NODE model recovered the dynamics of the winner-take-all system, including approximating the two distinct fixed-point attractors that correspond to the circuit's stable decision states (Fig. 3A). Critically, the model's interpretable structure allowed us to decompose the learned dynamics into the influences from each population at a given time (the terms of the decomposition in equation 8; Fig. 3). This analysis confirmed that the stable states emerged from a balance of strong self-excitation within the winning population and targeted inhibition from the shared I pool, where individual population influence arrows cancel out at fixed points (Fig. 3A bottom, colored arrows; Fig. 3B at the end of the trial). The early increase in activity is primarily driven by self-excitation, prior to the I population increasing enough to balance out this excitation (Fig. 3A top, colored arrows; Fig. 3B early in the trial). These results serve as a successful proof-of-concept, validating that our framework can accurately learn and expose the underlying mechanisms of a nonlinear neural dynamical system.

## 3.4 RESULTS: RELAXING MODEL CONSTRAINTS

In actual experiments, cell types of each recorded neuron are not always known. We thus tested the ability of our model to infer the cell type labels when only a fraction of the neurons' cell types were known. We modified the readout mask ($M_{pop}$ in equation 9) to have a block-diagonal structure with only two blocks, one for both E populations (E1 and E2) and one for the I population. This is because a researcher may only know E versus I, rather than knowing E1 or E2 ahead of time, as we previously assumed. We then unmasked the off-diagonal sections of the readout mask at various levels: 25%, 50%, and 75% (dashed boxes in Fig. 4) before training the CT-NODE model with an L1 penalty of $1e-4$ on the unmasked readout weights. Using this approach, we find that the CT-NODE model can accurately infer the cell-type identities of most of the unlabeled neurons. Additionally, the CT-NODE model learns to separate the E latents into E1 and E2 latents (versus using both E latents as a distributed code) without explicit instructions to do so (Fig. 4). When unlabeling half

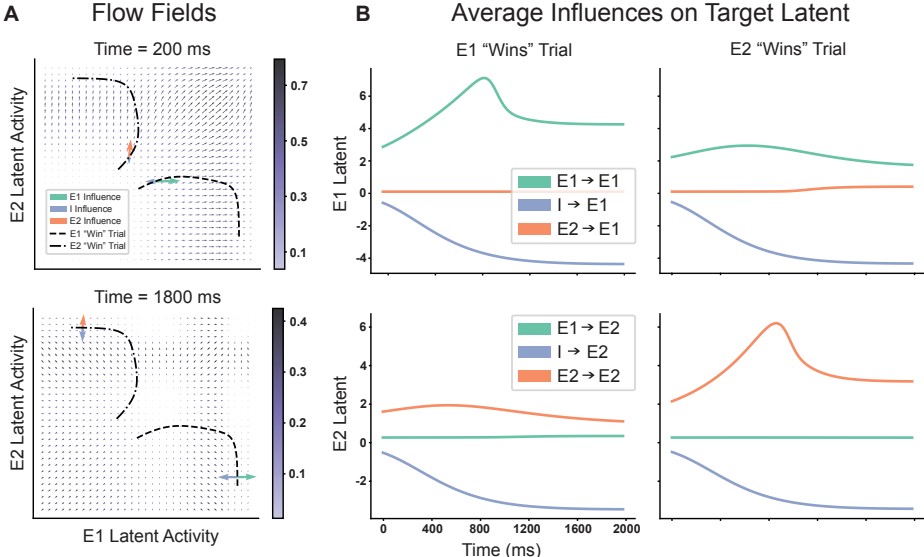

Figure 3: *Synthetic data dynamics interpretability*. **A)** Vector flow fields learned by CT-NODE of E1 vs. E2 latents at two different values of I, corresponding to average I values at two different time points early (top) and late (bottom) in the trial. Overlaid are two example trials where E1 and E2 'win' (trajectories to lower left and upper right, respecively). In colored arrows, we plot the contributions of individual populations towards the vector field at the listed times. Influence arrows at time 200 ms show strong, non-zero self-influence from the winning population. Influence arrows at 1800 ms show near fixed-point cancellation between the winning E and I population. **B)** The extent to which each latent population drives the flow field dynamics ('influences') of E1 (top row) and E2 (bottom row) latents. Columns separate trial averages by trial type.

of the E or I neurons, the model can identify the cell-types of these unidentified neurons with an accuracy of 98.0%. Even when only 25% are known, the model achieves an accuracy of 92.7%. This provides a proof of principle for the ability to apply CT-NODE models when not all cell classes are known, and additionally, to discover these unknown cell class identities.

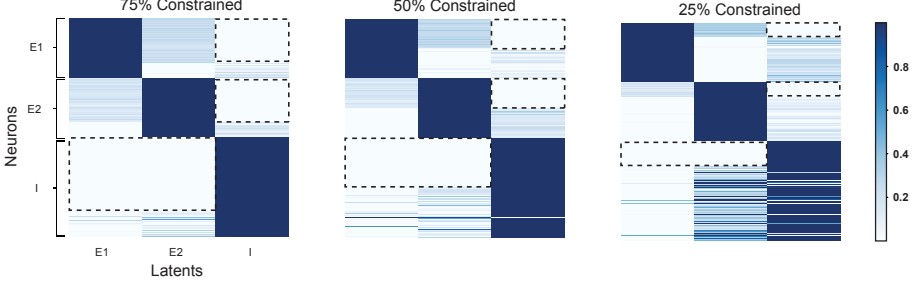

Figure 4: *Learning unlabeled cell types*. CT-NODE readout weights are shown across varying levels of constraints. Each row (neuron) is normalized to have a maximum value of 1 - thus, the color of each cell in a row denotes the relative amount each latent is used to predict a given neurons activity. Here, we only constrained a portion of the readout, under the assumption that only a portion of neurons' cell types were known. Within the dashed lines, the readout was forced to be 0, and all other values were learned (e.g. for the 50% constrained scenario, we enforced that 50% of the excitatory neurons could not map to the inhibitory latent, and that 50% of the inhibitory neurons could not map to the excitatory latents. Cell classes were generally inferred correctly by CT-NODE when using partial readout masks.

## 4 APPLICATION TO NEURAL RECORDINGS DURING NATURALISTIC CLIMBING

We next modeled datasets in which motor regions primary motor cortex (M1) and secondary motor cortex (M2) are simultaneously recorded in mice with Neuropixels probes, while they performed a naturalistic, self-paced climbing task (Kristl et al., 2025) (Fig. 5A).

### 4.1 MODEL FITTING DETAILS

We fit models to binned spiking activity of 189 neurons across regions M1 and M2 of a single mouse. Consistent with prior work, we classified neurons into I and E classes based on their waveform. With a threshold width of 13 ms, neurons above this threshold were classified as E and neurons below were classified as I (Kristl et al., 2025).

In total, our dataset contained four populations of neurons: M1-E (45 neurons), M1-I (19), M2-E (109), and M2-I (16). All spiking activity was binned into 20 ms bins and neurons with a firing rate less than 0.5 Hz across all trials were labeled as "silent" and removed. Models were trained and evaluated on a total of 227 trials of varying durations when the mouse was actively climbing. We randomly split the data by climbing trials, with 80% of the trials used for training and 20% for validation, making sure the split accounted for total climbing time (the training and validation set contained 80% and 20% of the total climbing time, respectively). To deal with the inconsistent trial lengths, each trial was split into 260 ms windows (13 bins) with 240 ms of overlap (12 bins). When inputting the windows into each model, we considered a context window of 200 ms (10 bins) for the initial condition encoder to causally predict a subsequent 60 ms (3 bins) horizon. We removed the GRU encoder's bidirectional property to reduce model complexity for causal training and evaluation. Population latent dimensionalities were sized to reflect the respective number of neurons in each neural population (15 for E and 6 for I populations, resulting in 42 latents total).

### 4.2 RESULTS: PREDICTION ACCURACY

As in the synthetic dataset, we compared the accuracy of our model with several other architectures (Fig. 5B). CT-NODE slightly outperformed a standard NODE, again demonstrating that the constrained dynamics architecture does not harm performance, and may even slightly benefit performance on held-out data. Interestingly, CT-NODE here had comparable performance to a NODE model with purely linear dynamics. We expect this may be due to the short decoding horizon that we used, which was important for making predictions in this naturalistic task with fairly inconsistent dynamics across climbing trials. CT-NODE also outperformed a model with RNN dynamics. In general, CT-NODE fared comparably or better than the performance of these other baselines.

### 4.3 RESULTS: DYNAMICS INTERPRETABILITY

The CT-NODE model architecture allows us to interpret the functional interactions between neural populations across regions. We compared how these functional interactions changed across two different types of behavioral epochs, those where the mouse was climbing, and those where the mouse was not (and was primarily still). Separate CT-NODE models were fit to these different behaviors, and we computed the average influence of each population on each other across time (Fig. 5C). As expected, we found that functional interactions were substantially stronger during climbing than non-climbing periods. Interestingly, during climbing, we found that the I influence from M1 to M2 was stronger than M2 to M1. (We emphasize that these are purely functional influences, as there are not anatomical I projections across regions). This modeling result could help to explain past surprising experimental results during climbing in which inactivating M2 led to a smaller effect in M1, relative to inactivating M1's influence on M2 (Kristl et al., 2025) (which is flipped from classical hierarchical descriptions of M2 impacting M1).

## 5 DISCUSSION

Here, we introduced Cell-type neural ordinary differential equation (CT-NODE) models, that enable modeling the interactions between E and I populations within a flexible nonlinear dynamical

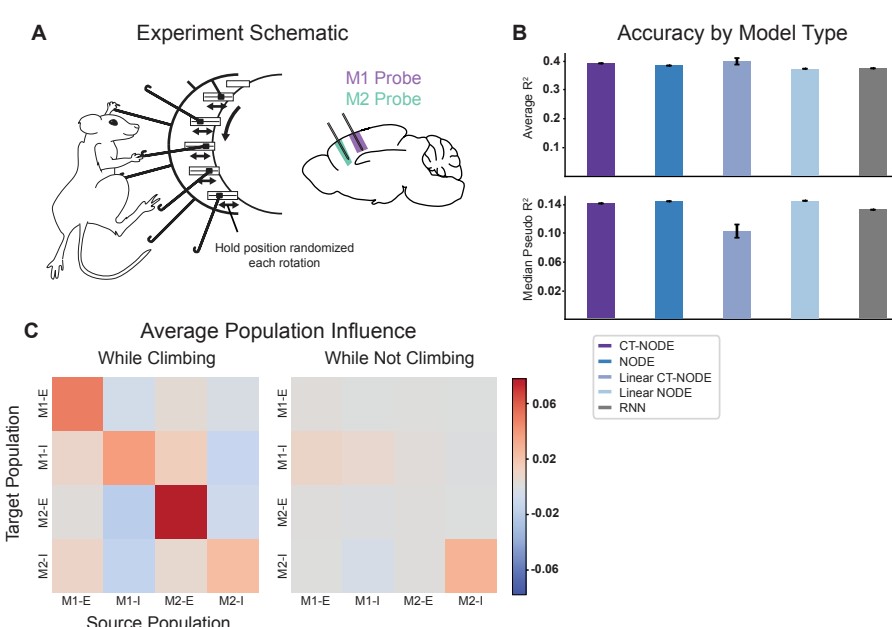

Figure 5: *Naturalistic Climbing Task Setup, Model Performance, and Interpretability.* **A)** Schematic describing mouse task. Probes inserted into M1 and M2 recorded neural activity during self-paced climbing. **B)** Future prediction accuracy is compared across model types. On top, the $R^2$ of the population averages. On bottom, the median pseudo-$R^2$ of individual neurons' spike count predictions. **C)** The average (over time) extent to which each population drives the others' dynamics (their 'influence'), for CT-NODE models trained on climbing and non-climbing data.

system. In synthetic and recorded Neuropixels probe datasets, we demonstrated how they retain the prediction accuracy of standard NODE models, while providing a newfound interpretability of how constituent populations drive the overall dynamics of a system.

A model limitation is assuming that we know the cell type identities of all neurons. Classifying neurons as E or I based on waveform characteristics, as we did with the recorded neural data, is an imperfect heuristic. While we showed initial results that we could learn cell types with partial knowledge - in particular if a portion of the neurons had known cell classes, it would be valuable for future work to extend this line of inquiry. For instance, to model the scenario where we are unsure about many cell types (e.g. due to imperfect waveform classification), we could couple L1 regularization for sparsity, with Tikhonov regularization to penalize readout weights inversely to the confidence we have about the classification.

Another central limitation of our current work is that the NODEs are fully deterministic in their dynamics and do not model noise in the underlying dynamics. Thus, the way that the model accounts for single-trial noise and variability is fully through the learned initial condition. The 'ground-truth' in the synthetic data is that the initial conditions for trials in which the two different E populations end up winning are actually almost identical - rather, a combination of input noise early on, coupled with the E/I dynamics, drives the model towards one solution. Using deterministic NODEs to best model this system, the models rather learned two separate locations of initial conditions for the trials in which the two E populations won. Thus, it would be valuable in future work to extend our CT-NODE framework to neural stochastic differential equations (NSDEs), which have successfully modeled neural activity (Kim et al., 2023). The dynamics architecture we developed should be readily interchangeable for standard NODE dynamics within NSDEs. In fact, our proposed dynamics architecture could more generally offer a path towards understanding cell-type-constrained interactions within a broad range of architectures of nonlinear dynamical systems modeling (Hernandez et al., 2018; Karniol-Tambour et al., 2022; Liu et al., 2025).

## 6 REPRODUCIBILITY STATEMENT

Details of all datasets and preprocessing steps are outlined in sections 3 and 4. The CT-Node details are outlined in section 2.3 and comparison model details are outlined in Appendix A. Anonymous source code is provided in the supplementary materials of the submission, and upon acceptance, the code and datasets will be publicly shared.

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

## A  COMPARISON MODELS

As stated in section 2.5, multiple dynamics models were trained to serve as comparison points to the CT-NODE model. The GRU-based encoder and linear mapping from hidden state $h$ to initial latent $z_0$ outlined in 3 was held constant across all model types. All models also contained a dropout layer with probability $p$ after the GRU encoder and before the linear mapping to the latents. Training hyperparameters were also kept constant between models, however specific hyperparameter values varied across tasks. All models were optimized with an Adam optimizer with a specified weight decay. The table below summarizes the hyperparameter differences across the simulated and climbing tasks the models were applied to.

| Hyperparameter Differences across Datasets | | |
|---|---|---|
| Hyperparameter | Simulated Task | Climbing Task |
| epochs | 3000 | 200 |
| learning rate | 0.001 | 0.0003 |
| batch size | 16 | 256 |
| weight decay | 0.0 | 1e-5 |
| dropout ($p$) | 0.05 | 0.3 |

The sections below outline any important differences between the model dynamics and/or readouts.

### A.1  LINEAR CT-NODE

As the name implies, the Linear CT-NODE is the linear version of the CT-NODE model. The dynamics of this model are described by 8, however the dynamics functions, $\tilde{f}_{jk}$, do not contain a nonlinear activation function and their outputs are not wrapped in a softplus activation. The readout is identical to the standard CT-NODE described by 9. The hidden dimensionalities of each monotonic neural network are the same as in the CT-NODE model with a dimensionality of 32 for the simulated task and 64 for the climbing task.

### A.2  UNCONSTRAINED NONLINEAR AND LINEAR NODEs

The dynamics of both the unconstrained nonlinear and linear NODEs are described by 1, where $f_\theta$ is parameterized by a two-layer neural network:

$$f_\theta(z_{t-1}) \;=\; W_2\phi(W_1 z_{t-1} + b_1) + b_2, \tag{11}$$

The activation function, $\phi$, is the $tanh$ nonlinear activation function for the Nonlinear NODE and the Identity function for the Linear NODE. Weight matrices $W_1$ and $W_2$ are not constrained to be

monotonic, as in the CT-NODE models. The two-layer neural network has a hidden dimensionality, $h$, set to 128 for both the simulated and climbing tasks. The readout of this model is described by 9, however, $W_{dec}$ is not constrained to be monotonic.

### A.3 STANDARD RNN

The dynamics of this model are described by 1, where $f_\theta$ is a standard RNN with no input.

$$f_\theta(z_{t-1}) = tanh(W_{rnn}z_{t-1} + b_{rnn}) \tag{12}$$

Again, the weight matrix $W_{rnn}$ and the readout weight matrix, $W_{dec}$ described by 9, are not constrained to be monotonic.

## B METRICS

### B.1 AVERAGE $R^2$ SCORE

The first metric to evaluate model reconstruction performance is the average $R^2$ score. This metric describes the goodness of fit between the average observed spikes and average predicted rates (as shown in Fig. 2C). It is computed by calculating the standard $R^2$ score between the mean spiking activity and mean predicted rates across all neurons within each population for each individual trial. The reported average $R^2$ scores in the Results sections are the mean scores across all trials and across all neural populations.

### B.2 PSEUDO $R^2$ SCORE

The Pseudo $R^2$ score is a goodness-of-fit metric that generalizes the standard $R^2$ to models with non-Gaussian response variables, such as the Poisson distribution commonly used for neural spike counts. It quantifies the model's performance by comparing the log-likelihood of its predictions to that of the mean spike count across an entire trial. We did this for individual neurons, for each trial, then averaged across neurons, giving us the average single-trial neuron pseudo-$R^2$.

