# OpenReview forum: "Cell-type Neural Ordinary Differential Equation Models for Parsing Biologically-Constrained Contributions to Neural Dynamics"
_ICLR.cc/2026/Conference — Submitted to ICLR 2026_

### Official Review · Reviewer_6bAV · 2025-10-28

**Soundness:** 2
**Presentation:** 2
**Contribution:** 2
**Rating:** 4
**Confidence:** 2

**Summary:**

The authors introduce CT-NODE, a framework for modeling neural population activity that incorporates biological constraints. CT-NODE extends standard NODE models by: 1) decomposing the overall dynamics into distinct, interpretable components that describe the interactions between different neural populations (e.g., cell types or brain regions) , and 2) enforcing Dale's law, which dictates that excitatory and inhibitory neurons have sign-constrained effects on other neurons. The authors validate the model on both a synthetic  and real neural recordings. They demonstrate that CT-NODE provides equivalent or better prediction accuracy compared to standard NODEs while enabling a new level of biologically-grounded interpretation of how different cell populations contribute to and shape neural dynamics.

**Strengths:**

- **Originality**: the work introduces biological constraints into nonlinear NODE. While the ideas has been previously incorporated into other architectures applying it to the NODEs setting is novel. This allows interpretability of the learnt dynamics, recovering, causal interactions between distinct neural populations.

- **Quality**: the paper sets suitable experiments to demonstrate the model allows for interpretability -- recovering underlying mechanisms.

- **Clarity**: the paper clearly presents the background, NODE, and motivation for the framework. Next the CT-NODE construction is easy to follow. At last, evaluations depict the core contribution, interpretability of the dynamics.

- **Significance**: though dependent on prior-population knowledge (cell classification at lease on a sub-set of the population); the method has great potential as it maintains predictive accuracy while allowing for interpretability.

**Weaknesses:**

- **Baselines for interpretability**: while the authors justify the contribution of the method is in including interpretability while preserving accuracy; comparisons are provided only for performance, evaluation wrt methods designed mult-population systems are missing (e.g. latent dynamics models), challenging the ability to properly assess the contribution.
- **Accuracy metrics**: though the authors accurately claim for "fared comparably or better than the performance of other baselines", to increase the significance of the paper it will be interesting to explore how one can actually improve on these or investigate why the biological priors do not assist in this task (prove the presented hypothesis of short prediction horizon or suggest an alternative explanation). With that it will also be valuable to provide quantitative performance values, as tables rather than bar-plots.
- **Additional biological insights**: given the prior comment, and that the contribution mostly lies in biological findings, it will be valuable to include additional examples (real/synthetic) of its value.
- **Acknowledged weaknesses**: the authors accurately describe existing limitations of the framework which will be valuable to address to increase its relevance --
(1) _deterministic dynamics_: the model's dynamics are deterministic; hence trial-to-trial variability in the neural activity is explained solely by different learned initial condition. This does not represent real, noise driven systems/measurements.
(2) _cell type labels_: the interpretability/population analysis requires prior knowledge, labels. For real measurements this information may not be available, making the method not applicable. Further, this introduces an inherent bias. One can consider a setting well classification is done based on measurements similarity, and is part of the learning framework.

**Questions:**

Following the above weaknesses:
1. Can the authors include a population dynamics baseline, and report evaluation on interpretability?
2. limited improvement in accuracy; can the authors explore this point beyond the presented hypothesis?
3. Demonstrate a biological finding unavailable from alternative models?
4. Follow the suggestions in the discussion; include stochastic ablations and re-visit the cell-label dependency.

---

### Official Review · Reviewer_YVyL · 2025-10-29

**Soundness:** 3
**Presentation:** 2
**Contribution:** 2
**Rating:** 2
**Confidence:** 4

**Summary:**

This paper introduces CT-NODE, an extension of neural ODEs that explicitly parameterizes subpopulations to obey biologically motivated constraints. To impose Dale's law, they assign separate latent states to excitatory and inhibitory neurons, and enforce the sign of cross-population influences by parameterizing them with monotonic neural networks. They evaluate their model on recordings from a synthetic winner take all task, arguing that their method yields good accuracy and gives more interpretible results. Lastly, they apply their method to neural recordings from mice performing a naturalistic climbing task, and use their fits to make predictions about how subpopulations of neurons influence each other during the task.

**Strengths:**

- Methods are clearly explained and easy to understand.
- CT-NODE yields fits that achieve decent reconstruction accuracy on both synthetic and real neural data.

**Weaknesses:**

The proposed method seems like a very incremental modification of existing methods; the novelty here is rather limited. The paper proposes something very similar to what was proposed by Jha et al. 2024, with the only substantial difference being the choice of neural ODEs as opposed to latent linear dynamical systems. While certainly more expressive, I disagree with the sentiment that these NODE-type models enjoy greater interpretibility. Rather, this class of models loses interpretibility in the sense that estimated connectivity/dynamics matrices are not readily extractable, whereas both flow fields and dynamics matrices can be directly inferred from LDS models (and their more expressive nonlinear variants like switching LDS).

Comparisons to other baseline models are only shown in the context of data reconstruction accuracy. However, no such comparisons are made for the overwhelming bulk of the presented results, such as cross-population influences interpreted by CT-NODE on both the synthetic and real neural data, as well as the cell type learning task. E.g. do other methods yield similar "interpretations", or does CT-NODE find a particularly unique/parsimonious interpretation of the data? Is CT-NODE especially good at cell type classification compared to other methods? As it stands, the argument for the merits of CT-NODE in terms of downstream interpretibility feels underdeveloped.

**Questions:**

See above.

---

### Official Review · Reviewer_sM9m · 2025-10-31

**Soundness:** 3
**Presentation:** 3
**Contribution:** 3
**Rating:** 6
**Confidence:** 4

**Summary:**

The authors extend the Neural ODE framework to account for multiple populations with Dale’s law. Specifically, they have a latent variable (of some small dimension) for each population. The effect of this latent on other latents obeys Dale’s law, by using monotonic neural networks and a softplus function. This architecture is test on a synthetic winner-take-all task, and on neuropixel data from rodents doing a climbing task.

**Strengths:**

Inference of dynamical systems from data is an important topic, with many methods suggested in recent years. The aspect of cell types has received relatively little attention within this field.

**Weaknesses:**

There are several controls that seem essential for understanding the robustness of the method. For instance, what happens if we guess the wrong number of cell types?

Presentation: many typos and small figures.

**Questions:**

Section 3.3 and figure 3: From the text it’s not clear whether the authors claim that the model inferred fixed points. Looking at the flow lines, it seems that there are no fixed points.
Lines 162-164 and Figure 1 – why unconstrained within population? This does not respect Dale’s law, which is a main claim in the paper. If I understand correctly, in the paper by Jha et al, the single-latent diagonal is unconstrained. In the current paper, it seems the entire diagonal block is unconstrained.
Small typos: line 115 specific – specified. 116 matter – manner. And many more throughout the paper.
Line 139 – why is only the self-term nonlinear?
Figure 2B – are the error bars standard deviation, or standard error of the mean? What is N?
Winner take all example: Do you have to know the number of blocks a priori? What happens if you guess wrong?
Neuropixel example: How robust are the results to the choice of waveform threshold?
Spike waveform threshold: In the Kristl paper this was 0.417 msec. Here it is 13msec.

---

### Official Review · Reviewer_CRvo · 2025-11-01

**Soundness:** 3
**Presentation:** 3
**Contribution:** 3
**Rating:** 2
**Confidence:** 3

**Summary:**

This paper introduces the Cell-Type NODE (CT-NODE) model, a neural ordinary differential equation framework that incorporates Dale's law constraints to model excitatory and inhibitory neural population dynamics. The authors decompose overall dynamics into population-specific components using monotonic neural networks to enforce sign constraints on cross-population interactions, while leaving the intra-population interactions unconstrained. The approach is validated on synthetic winner-take-all data and real neural recordings from the mouse motor cortex during climbing behavior.

**Strengths:**

There are some interesting contributions to the field, such as:
1.	Incorporating Dale’s law constraints into latent interactions via NODE is a valuable direction for interpretable population models.
2.	Using monotonic neural networks with softplus wrapping (Eq. 6) to enforce sign constraints is technically sound.
3.	The synthetic data simulation demonstrates that the CT-NODE can have similar or better performance than other methods and can provide additional interpretability.

**Weaknesses:**

Several critical issues significantly limit the contribution:

1.	The authors state they draw inspiration from Cell-type Latent Dynamical Systems (CTLDS; Jha et al., 2024) for incorporating E/I constraints.  However, a key difference between CTDS and CT-NODE is that CT-NODE is a population-level model, where the latents represent an E or I population. The author constrained only cross-population interactions and left the intra-population interactions free. This design choice lacks biological and theoretical justification. I recommend checking the classical Wilson-Cowan E/I population model and its modern learnable extensions to compare and justify their model.

2.	The authors mentioned they trained two types of models: continuous integration and a specific discretized version with a timestep 0.1. However, no results are presented in the main text or the appendix to show that these two models have equivalent performance as they claimed.

3.	Insufficient evaluation of dynamics quality: for synthetic data, the metrics only focus on spike prediction accuracy, but fails to validate the quality of learned dynamics: no validation that the learned flow fields match ground truth dynamics. Missing ablations: 1). What is the effect of latent dimensionality on both performance and interpretability 2). Comparison with equal parameter-count baseline models.

4.	Figure 5 shows the real data results, but it is significantly incomplete compared to Figure 2 of the synthetic data simulation. Also, for this task, the authors use only a 60ms (3-bin) prediction horizon, which is far too short to assess dynamical modeling. Moreover, there is no visualization for how the prediction performance and population-specific metrics are.

**Questions:**

Same as weaknesses above.

---

> ### Author Response · Authors · 2025-12-01
> **Thanks to all reviewers**
>
> We know that discussion is no longer possible at this point, but we just wanted to thank all the reviewers for their extremely helpful and thorough comments.
>
> Most notably (and not exclusively), the comments have:
>
> -Driven us to more thoroughly investigate the benefits of the dynamics interpretation of our nonlinear CT-NODE model versus a linear model (as in the previous CTDS work). We have found that linear models are not able to discover as close to a fixed-point structure in the synthetic datasets, and this leads to influences between E and I populations not being accurate.
>
> -Motivated us to explore longer time-horizon predictions in the real neural activity dataset.
>
> -Helped us to correct our model formulation, which was previously written incorrectly, so that only self-terms for individual latents (not entire populations) are unconstrained.
>
> Thanks again!

---

### Meta-Review · Area_Chair_Ebrz · 2026-01-09

**Summary:**

This paper introduces CT-NODE, a neural ordinary differential equation framework that incorporates biologically motivated constraints, specifically Dale’s law, to model excitatory and inhibitory neural population dynamics. The reviewers recognize that incorporating cell-type structure and sign constraints into nonlinear dynamical models is a meaningful direction, and note that the proposed use of monotonic neural networks to enforce cross-population sign constraints is of interest.

Despite these strengths, reviewers raise substantial concerns about the conceptual justification and rigor of the evaluations. Reviewer CRvo highlights a key modeling choice that lacks biological justification. Reviewer sM9m echoes this concern and questioned why entire diagonal blocks are unconstrained. They raise further questions about whether the learned dynamics actually exhibit fixed points. Reviewer YVyL further argues that the method is an incremental extension of existing approaches, differing primarily by replacing latent linear dynamics with NODEs, and questions whether this shift genuinely improves interpretability relative to established latent dynamical systems.

The experimental evaluation is viewed as insufficient to support the paper’s interpretability claims. Reviewers CRvo, YVyL, and 6bAV note that comparisons to baselines focus almost exclusively on reconstruction accuracy, with little evidence that CT-NODE yields uniquely informative or superior interpretations of population interactions compared to existing models. Important controls and ablations are missing, including sensitivity to latent dimensionality, robustness to incorrect assumptions about the number of cell types (sM9m), validation of learned flow fields against ground truth dynamics (CRvo), and evaluation over longer prediction horizons in real data.

While the authors submitted a brief closing comment thanking reviewers and noting intended future investigations and corrections, they did not provide a point-by-point rebuttal or engage directly with the reviewers. As a result, key concerns regarding biological justification, interpretability claims, and rigor remain unresolved. Taken together, the reviewers’ feedback indicates that the work is not yet ready for publication.

**Reviewer Concerns:**

The concerns about the assumptions underlying the model, evaluations, and justification of the interpretability claims were not resolved during the rebuttal period.

**Reviewer Scores:**

I don't think the reviewers would have changed their scores.

---

### Decision · Program_Chairs · 2026-01-26

Reject